# Design and On-Field Validation of an Embedded System for Monitoring Second-Life Electric Vehicle Lithium-Ion Batteries

**DOI:** 10.3390/s22176376

**Published:** 2022-08-24

**Authors:** Diego Hilario Castillo-Martínez, Adolfo Josué Rodríguez-Rodríguez, Adrian Soto, Alberto Berrueta, David Tomás Vargas-Requena, Ignacio R. Matias, Pablo Sanchis, Alfredo Ursúa, Wenceslao Eduardo Rodríguez-Rodríguez

**Affiliations:** 1Department of Computational Sciences and Technologies, Computational Systems Academy, Autonomous University of Tamaulipas (UAT), Reynosa Rodhe Multidisciplinary Academic Unit, Reynosa-San Fernando Highway, Reynosa 88779, Tamaulipas, Mexico; 2Institute of Smart Cities (ISC), Department of Electrical, Electronic and Communications Engineering, Public University of Navarre (UPNA), Campus de Arrosadia, 31006 Pamplona, Spain

**Keywords:** Li-ion battery, stationary energy, storage system, embedded system

## Abstract

In the last few years, the growing demand for electric vehicles (EVs) in the transportation sector has contributed to the increased use of electric rechargeable batteries. At present, lithium-ion (Li-ion) batteries are the most commonly used in electric vehicles. Although once their storage capacity has dropped to below 80–70% it is no longer possible to use these batteries in EVs, it is feasible to use them in second-life applications as stationary energy storage systems. The purpose of this study is to present an embedded system that allows a Nissan^®^ LEAF Li-ion battery to communicate with an Ingecon^®^ Sun Storage 1Play inverter, for control and monitoring purposes. The prototype was developed using an Arduino^®^ microcontroller and a graphical user interface (GUI) on LabVIEW^®^. The experimental tests have allowed us to determine the feasibility of using Li-ion battery packs (BPs) coming from the automotive sector with an inverter with no need for a prior disassembly and rebuilding process. Furthermore, this research presents a programming and hardware methodology for the development of the embedded systems focused on second-life electric vehicle Li-ion batteries. One second-life battery pack coming from a Nissan^®^ Leaf and aged under real driving conditions was integrated into a residential microgrid serving as an energy storage system (ESS).

## 1. Introduction

Today, fossil fuel-based power sources are contributing to increased environmental pollution and causing damage to human health and ecosystems. The need for transportation in cities is progressively rising due to the growing population and to industrial activities, leading to increased greenhouse gas emissions. Vehicle traffic accounts for 74% of total transport energy consumption [1]. Renewable sources based on wind and solar energy [2,3] stand as an alternative to produce clean power, while electric vehicles powered by green energy offer an opportunity for sustainable growth [4,5]. The EVs electrochemical energy storage systems are mainly based on Li-ion batteries. Thanks to their high energy and power density compared to other technologies, these devices are preferred by automotive manufacturers. Unfortunately, the life cycle decreases as a result of the continuous charge/discharge processes, reaching an end-of-life state when its storage capacity drops to around 70–80% [6], requiring replacement with a new one. The retired Li-ion batteries are then destined to be either disposed of, dismantled for recovery of their mineral constituents, or repurposed in a second-life application [7]. To help reduce the investment costs and environmental pollution, the ideal choice is to repurpose the Li-ion batteries for a second life in less demanding stationary applications that do not require performance at full energy storage capacity [8], such as residential, industrial, and renewable systems [9]. Studies confirm such EV Li-ion battery second-life applications using batteries manufactured by BMW^®^, Daimler^®^, and Nissan^®^ [10]. To achieve the mentioned applications, some methods rely on the disassembly of the internal cells in the BPs, which is an expensive and slow task. The interest of this present study lies in the fact that there is no need to disassemble the Li-ion batteries, demonstrating this with a Nissan Leaf^®^ first generation EV BP. One of the main devices of the Li-ion batteries is the battery management system (BMS), which are responsible for the measurement of the *battery operating parameters* and *tasks*, such as the voltage current flow and temperature (VCT), stability of charge, ON/OFF switch, charge/discharge, time operation, and alerts. It also monitors battery states such as *state of available power* (SOP), *state of charge* (SOC), *state of life* (SOL), and *state of health* (SOH) [11]. It is important to consider that the BMS was not originally designed to work with photovoltaic (PV) inverters as a stationary ESS. Therefore, there is a need to design a device based on hardware and software that creates communication between the inverter and the Li-ion BMS in order to control and monitor the ESS. 

The state-of-the-art literature provides knowledge of studies focused on the design and development of technologies of this kind. There are studies that put forward systems based on microcontroller technologies that permit communication with the EV BMS. *L. Yuheng et al.*, *A. Rahman et al.*, *P. Yang et al.*, and *S. M. Salamati et al.* designed devices to determine the battery VCT through communication with the BMS using the NXP^®^ MC9S08DZ60, Arduino UNO^®^ ATmega328 and Infineon^®^ MCX2365B microcontrollers, and Raspberry Pi^®^ microprocessors, respectively [12,13,14,15]. In the previous proposals there is not a GUI. The GUI act as a human–machine interface (HMI), providing the user with real-time knowledge of the BMS operating variables and tasks. Furthermore, its implementation is very important in order to collect the information provided by the BMS regarding the state of safety, usage, performance, and longevity of the battery. This is because, in the event of undesirable circumstances such as overcharging/heating, these must be detected and corrective actions must be performed automatically through programming systems [11,16]. Using the same technology M.H. Abd-Wahab et al. reported the development of an interface based on an Arduino UNO^®^ ATmega328 only monitoring the voltage of the Li-ion battery using a sensor, and a GUI for monitoring the system via a web page [17]. Following the strategy based on the microcontrollers Arduino UNO^®^ ATmega328 and ESP32^®^, *A. Jamaluddin et al.* and *M. H. Qahtan et al.* presented systems to the battery voltage and current-flow monitoring by sensors and GUIs realized on LabVIEW^®^ and Blynk^®^ [18,19]. On the other hand, some studies have demonstrated the experimental setup processing by Data Acquisition (DAQ) modules and programmable logic control (PLC). *E. Soylu et al.* reported an experimental setup based on the Advantech^®^ USB-4716 DAQ to measure the battery VCT and a GUI developed by Visual Studio^®^ 2010 [20]. *E. Locorotondo et al.* showed a test setup constituted by the Dspace^®^ MicrolabBox DS1202 DAQ applied to VCT sensing controlled by MatLAB^®^ Simulink [21]. *J. Fleming et al*. designed an electromechanical and thermal monitoring via PLC and a Windows application based on C# to visualize the Li-ion battery data as HMI, establishing communication with the measurement system utilizing the anode and cathode connection within the cell for data transmission, i.e., connecting the experimental setup across the electrical battery terminals [22]. The BMS must communicate with the on-board management system, which controls all the vehicle’s parameters, sending status information and receiving messages and alerts, such communication tasks are performed by the controller area network (CAN) [23]. The BMS uses the CAN protocol to send data relating to the operating parameters and required tasks for the Li-ion battery performance. *Ch. Hommalai et al.* reported an experimental setup based on the CAN protocol to establish communication between the Li-ion battery and a PLC applied to detect dead battery cells, where the VCT readings are displayed by a GUI [24]. Nevertheless, besides the VCT acquisition in order to design a battery monitoring system, it is relevant to obtain the BMS messages. These messages allow to achieve knowledge about real-time battery charge data, such as *full charge, remaining capacity, new full capacity, remaining charge time, charge power status, charge and discharge power limit, maximum power charge*. In this article, we propose an experimental setup based on a hardware architecture capable to acquire these battery data charge through CAN communication with the BMS and a GUI that visualizes them. *H. Wang et al.* designed an interface based on the Texas Instruments^®^ MSP430 microcontroller acquiring VCT variables via CAN without the implementation of a GUI as HMI [25]. In addition to this present study, the state-of-the-art literature shows studies based on the Li-ion battery implementation from a Nissan Leaf^®^ automobile applied as a stationary ESS, presented by *M.T. Smith et al.* and *E. Braco et al.* [10,26,27]. There are even reports that expose the Nissan Leal^®^ Li-ion batteries second-life applications as ESSs on business models by companies such as 4R Energy^®^ in Yokohama, which offers commercial energy storage services [28], and FreeWire^®^ in California, which provides a mobile and portable charging station to EVs [29].

Considering the set-up of a stationary ESS based on a Li-ion battery and a PV inverter, we propose the design and development of a practical plug-and-play technology that is programmable and able to perform at any time in order to verify the current data, ready to act automatically in the presence of faults, and incorporates an HMI. With this aim in mind, we designed an interface based on an embedded system. The embedded systems are computer architectures based on microcontrollers/microprocessors energized by an internal or external power source, able to control actuators, acquire data from sensors through electric circuits, and integrate communication peripherals into the study process and into the GUI as an HMI. We therefore propose an embedded system to control and monitor the Li-ion battery operating parameters (VCT and SOC) and tasks (charge and discharge) when working with an INGECON Sun Storage 1Play 3TL/6TL inverter to manage an energy storage system for a residential microgrid. The proposed system comprises an Arduino MEGA^®^ ATmega2560 microcontroller and a GUI based on LabVIEW^®^. Our system has presented has been implemented with low-cost devices. The study is structured as follows. Section 2 explains the theoretical principles of Li-ion BPs, inverters, and embedded systems. Section 3 presents the materials and methods. Section 4 shows the experimental design of the software and hardware embedded in the system. Section 5 describes the experimental results obtained when the system developed is tested in a micro-grid to characterize its stability. Finally, the main conclusions are discussed in Section 6.

## 2. Theoretical Background

### 2.1. EV Li-Ion Batteries Second-Life Applications

Nowadays, Li-ion batteries are used in a range of applications such as those for aerospace, spaceflights, drones, automotive applications, and grid storage [22]. The battery lifetime is reduced by constant and prolonged use, heavy use, or harsh temperature conditions [24]. Charge and discharge operations lead to the degradation of the energy storage capacity of these batteries. In the automotive industry, a drop of more than 20% in the nominal capacity leads to the non-optimal performance of the EV. Considering the environmental and economic impacts, it is recommendable to re-use Li-ion batteries in ESS in residential, industrial, or renewable applications. Before repurposing a Li-ion battery as a stationary ESS, it is first necessary to verify its remaining capacity and internal health. A technical diagnosis also needs to be made [10,30]. At present, these batteries are extremely popular due to their long life cycle, high voltage performance, and low self-discharge. For this reason, they are well suited to ESS applications. The current Li-ion batteries must be controlled and supervised correctly with a device to ensure that each cell works within the VCT operating ranges, and the device that performs these operations is the BMS [31].

### 2.2. Battery Management System: BMS

The BMS is the central unit inside the Li-ion BP, responsible for acquiring the VCT data and controlling the actuators. Its principal task is to ensure that the battery is operating within safe limits and that it achieves optimal performance over its useful life under various operating and environmental conditions. Specifically, it needs to keep the temperature stable and the electrical operating variables within the predetermined limits to ensure that the system delivers the energy demanded at any time and to guarantee the health of the Li-ion battery [31,32]. According to *Y. Xing et al.*, a BMS for an EV comprises the following stages: *battery management* (user interface, electrical control, thermal management, and communication), *battery state* (state determination and safety protection), and *battery monitoring* (VCT data acquisition) [16], see Figure 1. A BMS comprises sensors, actuators, controllers, and communication interfaces.

Using the CAN communication protocol, the BMS communicates with the vehicle management system (VMS), which controls all the vehicle parameters and processes. Each car manufacturer defines certain messages to exchange communication between the BMS and the VMS through this protocol [23]. Due to the distinct design, it is mandatory to design and implement an interface between the Li-ion battery and PV inverter that interprets the messages from the BMS and establishes two-way communication. Furthermore, it must perform as an HMI via programming sequences and a graphical user interface. This can be achieved by an embedded system. 

### 2.3. Embedded Systems

Embedded systems (ESs) are an efficient technology strategy to perform dedicated functions supported by a computer architecture based on two general sections: *hardware* and *software*, permitting data acquisition and processing. The microprocessor (μP)/microcontroller (μC) is the core data processing unit, responsible for taking the control decisions to manage one or several processes. It is the main ES hardware element. The hardware comprises a μP/μC interconnection with electric circuits (drive circuitry and signal conditioning), actuators, sensors, and communication peripherals; i.e., the physical components. In the μP/μC the programmer writes the machine language coding through integrated development environment (IDE) software. The software is a set of instructions (called codes) written for the μP/μC to perform tasks called programs. The codes are based on programming sentences to achieve the management of its own hardware, which is able to control the study processes via communication interfaces (wired or wireless). These codes and data from the study processes are stored and entered, respectively, in the μP/μC memory [33]. As long as the programming IDE software impacts the programming directed at operator/administrator monitoring, it is possible to use a different software to develop the GUI for user operation [16]. The general architecture of an ES is illustrated in Figure 2. To summarize, *P. Zhan et al.* defines an ES as “*a computer based on hardware and software, which both, are physically embedded within a large industrial process system. Its mission is to ensure the communication and control over the study processes components, in order to achieve the overall system management*” [33]. Therefore, as indicated in the explanation above, a BMS is an ES directed at controlling the EV energy.

## 3. Materials and Methods

Regarding the electric vehicle Li-ion battery pack characterization and its second-life application to stationary energy storage, in this section we explain the materials and methods required to design our ES prototype. This device is capable of establishing communication between the Nissan^®^ EV First Generation Pack and the Ingeteam^®^ Ingecon Storage 1 Play inverter. The BP comes from an EV in which the driving conditions and the current state of the battery are unknown. Therefore, a characterization test was performed before its utilization. The measured parameters are current capacity, energy, energy efficiency, and coulombic efficiency. Moreover, the SOH is calculated as the ratio between the current capacity and the nominal capacity. With that aim, the battery was charged and discharged three times. The charging sequence consists of a constant current phase at 22 A (C/3) followed by a constant voltage phase until a cut-off current of 2.64 A (C/25) was reached. The discharge protocol consists of a constant current phase at 22 A between the maximum and minimum voltage limits. The characterization test was conducted at a room temperature of 22 ± 3 °C. Table 1 compiles the main parameters defined by the manufacturer and the mean value of the parameters measured on the test. 

Figure 3 shows the Arduino Mega^®^ ATmega2560 as the core system which, through communication peripherals, interprets the messages sent by the BMS to be transmitted to the inverter and monitored by the GUI. Therefore, the ES hardware acts as a liaison device (LD).

The LabVIEW^®^ GUI determines the actions to be executed over the virtual power controls and interprets the messages received via the indicators. Communication between the LD, the battery, and the inverter is two-way via CAN protocol. The communication via CAN protocol was implemented using the CAN-BUS Shield v1.2, which is based on the CAN MCP251 and MCP2551 controllers, and an SPI interface. It is important to consider power actuators to achieve the μC control over the battery as these are necessary to activate the BP internal relays. An Arduino^®^ 4-channel relay module was employed to do so. The data transmission between the Arduino^®^ and the LabVIEW^®^ GUI is made through a serial port.

## 4. Embedded System Design for Monitoring Second-Life EV Li-Ion Batteries

In this section, we explain the ES design and development for monitoring second-life EV Li-ion batteries integrated into a stationary ESS. This is carried out by a software- and hardware-based methodology defined to perform the experimental tests and the complete system implementation in a residential microgrid in order to achieve its characterization. In the interest of establishing communication between the ES via the Arduino^®^ Mega ATmega 2560 μC to control and monitor the battery operation towards the overall energy storage with an Ingecon^®^ Sun Storage inverter, it is necessary to manage the charge and discharge processes. The power of the inverter depends on the power balance of the microgrid, the state of the battery, and the strategy, as later discussed in Section 5. Therefore, active communication between the BMS, the ES, and the inverter is required. The inverter establishes a communication channel via the LD and its internal commands with the BP. Once this has been achieved, the GUI shows the operator or administrator the system performance. 

### 4.1. Communication Processes between the BMS and Li-Ion Battery

Firstly, one of the requirements to address when the BP is removed from the EV and reused is to establish the communication with the BMS by the LD. To realize this task is mandatory to know that each manufacturer establishes its own internal communication messages and processes. The Nissan^®^ EV first generation BP includes three processes: activation, charge/discharge, and deactivation. The μC interprets these processes as a translator and sends them to the inverter. Thanks to this approach, the Li-ion battery retired from an EV can be connected to a stationary inverter, thereby allowing its second use in a less demanding application in terms of power and energy density. Table 2 describes the steps required to address the three processes.

### 4.2. Second-Life EV Li-Ion Battery Work Sequence

This subsection begins by considering that, the battery originally is not designed to communicate directly with the inverter. Therefore, we propose a *work sequence* capable of managing and establishing the communication between the Li-ion battery and the inverter through the plug-and-play LD for a second-life stationary ESS application described by the flow diagram depicted in Figure 4. The work sequence proposed is based on the battery operational processes described previously. The first task is to verify the *battery activation process* by the operator through the GUI. When this occurs, it is possible to initiate the communication between the battery with the inverter through the. Once the communication is established, the inverter recognizes the battery operating parameters such as VCT, SOC, and messages. The second task to attend is the *battery charge/discharge process*, which depends on the operating parameters, and the operator decisions. The SOC value shows the usable charge capacity of the battery as a percentage. Afterward, the operators can execute the *battery deactivation process* whenever they want, disabling the battery, inverter, and ES power supplies. It is worth mentioning that the GUI notifies the operators and automatically deactivates the ESS if any hazardous electrical or thermal battery situations occur during charging or discharging processes.

### 4.3. Logical Design of the ES Operation

This section describes the embedded system’s architecture that constitutes the second-life ESS. In Figure 5, this is explained by a flowchart denoting the activation, charge/discharge and deactivation processes, and the involved parts tasks such as the operator (blue), GUI (green), LD (yellow), battery BMS (gray), and inverter (orange) and their interactions. The operator through the GUI is capable of visualizing and controlling the processes, knowing in real-time the battery operational variables (VCT, SOC, and messages). The execution starts with the activation process when the operator makes the request via the GUI. The LD programming is based on the sequential structure allowing to execute a series of instructions in order. Firstly, the LD initializes the communication with the BMS and the inverter, awaiting their activation confirmation. After that, when is confirmed the activation of both devices, the next stage can be performed. The charge/discharge process is available when the battery sends the operational parameters and alerts to the μC, acting as an information transmitter and emitter for the inverter. The GUI informs the operator in real-time about the current state of operational parameters. Based on that information, the operator decides which task to execute, either charge or discharge, if and only if no alerts are coming from the BMS. In real-time, the operator is informed by the GUI about the operational parameters’ current state and decides which task to realize charge or discharge, considering that there are not any alerts present from the BMS. These alerts, shown to the operator through the GUI, can be caused by low or high voltage/current flow readings or the presence of overheating. The operator receives real-time information about the current state and based on those can decide whether to start. The current process follows a repetitive cycle until the operator makes a new demand. Finally, the deactivation process occurs when the operator sends the request via the GUI, leading to the battery high-voltage output being turned off.

### 4.4. Liaison Plug-and-Play Device

This section describes the plug-and-play LD, which allows acquiring data directly from the BMS without needing external devices or sensors. This device has been designed based on the manufacturer’s technical specifications. Figure 6 shows the LD hardware architecture. Figure 7 depicts the developed interface, indicating its components. 

The electrical socket provides power to the internal power supply (i.e., RT-65D AC-DC 24 V), which transforms the outlet voltage level to one compatible with the liaison interface (e.g., 5 V). The communication between the μC, the BMS, and the inverter is made with two connectors: RJ-45 and 8-pin, respectively. Furthermore, the interface includes a start/stop button and 5 LEDs to indicate the current status, namely, normal operation (green) or a parameter that is out of limit (green), fault (red), warning (yellow), or beginning protocol (yellow). The Arduino^®^ Mega ATmega2560 achieves communication through two CAN-BUS Shields v.1.2. The LD communication task starts when the BMS sends the operating data such as VCT, SOC, and messages to the μC. Then, this information is processed by the Arduino to be sent to the inverter based on the CAN-BUS protocol established by each manufacturer. It is important to mention that both devices, the inverter and BMS, have distinct baud rates, so the μC needs to act as a buffer. To control the activation and deactivation processes, an Arduino 4 Relay Board is included to allow the LD control over the BMS. Furthermore, one of the relays is applied to control an external high voltage (HV) contactor (Omron G9EC1, 400 V, and 200 A DC). This HV contactor is able to interrupt the current flow if an alarm or error message occurs. For safety reasons, this protection system is based on two physically decoupled circuits. One is formed by the relay and controlled by Arduino through a power supply that does not exceed 24 V (located in the BP), and the other is formed by the HV contactor and controlled by the protective relay. To avoid noise coupling and false triggers, the HV contactor is located outside the BP. It is important to mention that the electrical sensing is not made by additional sensors or another experimental setup, and the electrical measurements are obtained via the BMS data. This is mentioned due to the fact that other studies, such as [12,13,14,15,16], have required external devices to obtain the electrical variables. 

### 4.5. Graphical User Interface-GUI

This section describes the GUI and its functionalities. The GUI is able to receive the data sent by the hardware interface and the battery through a LabVIEW^®^ virtual instrument. The user can control and monitor the activation, deactivation and battery charge/discharge processes through the GUI. The two-way communication from the hardware interface to the GUI relies on a USB cable. Figure 8 shows the GUI. Controls and indicators are grouped according to the message received. For reasons of confidentiality, it is not possible for us to mention the message names. For practical purposes these are named sections A, B, C, D, and E. The GUI is divided into the following sections:The *serial port control* allows the user to select the communication port through which the μC is connected to the PC, while the *stop button* reestablishes the connection and stops data acquisition and display.The *interface control* section allows the user to switch the hardware-interface ON/OFF. This section also shows the BMS and inverter connection status and if they are currently communicating with the μC.The *serial port data* section shows the messages received and its sole function is to display current activity.*Message section A* displays real-time battery data such as voltage and current flow, relay cut requests, main relay on, full charge, interlock, discharge power status.*Message section B* monitors real-time battery data such as remaining capacity, new full capacity, remaining capacity segment, remaining capacity segment switch, SOC, average temperature, output power limit reason, and remaining charge time.*Message section C* displays real-time battery data such as switch flag, high/low voltage times, temperature, wakeup phase, integrated current, cell voltage, state of health, and DTC, which is a variable with battery diagnosis information.*Message section D* displays real-time battery data such as SOC, IR sensor wave voltage, ALU answer (a diagnosis register for the CAN communication), IR sensor Malf (an alarm triggered if the insulation resistance sensor is malfunctioning), capacity empty, and refuse to sleep. *Message section E* monitors the real-time battery charge/discharge process data such as discharge power limit, charge power limit, charge power status, maximum power charge, and battery pack maximum UPRATE.The *flags* section uses virtual LED indicators to monitor the charge and discharge status such as overcharge, high voltage, high current, stop requests, over discharge, low voltage, and high current as well as the general battery status, as follows: high temperature, insulation resistance, CAN communication error, and unavailable values. Finally, the *flags* section mentions the current status of the relays. 

It is important to deploy the flags section in the GUI as this allows the operator to monitor the battery voltage range during the charge and discharge processes. When the battery voltage is below 124 V the low voltage, over-discharge, and stop request indicators are activated, and the system is disactivated. When the voltage is between 215 V and 274 V, this represents a warning state and the low voltage and stop request alert indicators are activated. When at the recommended voltage operating range from 275 V to 400 V, there are no alerts or errors. When the voltage is over 400 V to 407 V the high voltage and stop request alert indicators are activated. Finally, when the voltage is higher than 408 V, the overcharge and stop request are activated, so the system is disactivated.

Figure 8 depicts the GUI for real-time monitoring and control of BMS inverters, focusing on a battery discharge test considering a current-flow constant. The *serial port control* section visualizes the USB acquisition data by the PC Port COM4. The *interface control* section ensures that the BMS and inverter are connected to the LD; i.e., both devices are activated. The *serial port data* section shows the messages received by the BMS interpreted by the following sections. The *message section A* displays the operational battery voltage 371 V meaning that, this variable is inside from the recommended voltage operating range. The discharge battery current flow is −20 A. Besides, the *message section D* visualizes the SOC at 96%. Therefore, there are no alerts or fails in the *flags* section. The *failsafe status* and *discharge power status* equal the *relay cut request,* and the *main RLY* manifests the BMS inverter GUI’s correct performance.

## 5. Results

In this section, we present the real-time experimental tests performed on our ES applied to manage the energy storage for a residential microgrid. These tests were conducted at the Laboratory for Energy Storage and Microgrids of the Public University of Navarre. The ES was tested to establish the communication between the Nissan^®^ EV First Generation Pack and the Ingeteam^®^ Ingecon Storage 1 Play inverter. To assess the robustness and reliability of the device described above, the battery and the ES were integrated into a real microgrid, and the implementation is shown in Figure 9. A collective self-consumption scenario was emulated in the microgrid of the Public University of Navarre in northern Spain. To do so, the power consumption of four houses located in the vicinity of the university was monitored. Apart from the inverter, the microgrid includes a photovoltaic array of 11.5 kWp, and a power management system (PMS). 

The PMS is responsible for monitoring all the variables and setting the inverter power setpoint, which depends on the strategy implemented and the variables measured. In addition to the variables monitored by the microgrid, a Yokogawa WT-1800 power analyzer was included for the accurate measurement of the battery current and voltage. In the case of the results presented in this paper, the strategy aimed to maximize self-consumption. The energy is banked in the battery when the PV generation exceeds the consumption. Afterward, the battery is discharged when the PV generation does not meet the consumption. More information concerning the microgrid, boundary conditions, and the strategy can be found in a previous publication [34]. As an example to demonstrate the correct performance of the ES constituted by its hardware and software, Figure 10a shows the energy balance of the microgrid over 5 days, with a 5 min resolution. The yellow area corresponds to the energy consumed directly from the PV generation, whereas the orange area is the one used to charge the battery. The green area represents the energy delivered by the battery. Finally, the grey area is the energy consumed from the public grid. The energy consumption of the houses is represented by a solid black line. The integration of the ESS into the microgrid made it possible, on the one hand, to reduce the peak power absorbed from the grid and, on the other hand, to reduce the energy consumption from the grid, maximizing self-consumption. Table 3 compares the most relevant factors for a self-consumption installation: (1) peak power required from the grid, (2) energy consumed from the grid, and (3) self-consumption ratio. From Table 3, some of the advantages of including an ESS into a microgrid can be drawn. However, it is not the objective of this article to conduct a techno-economical analysis of the benefits of integrating a BP but to prove the feasibility of the developed ES. Figure 10b shows the Li-ion battery current and voltage throughout the 5 days of testing. A current positive sign implies battery charging, whereas a negative sign is set for discharge. The ES operated as expected for several weeks, demonstrating its suitability for stationary applications. The on-field validation was intentionally interrupted since a new strategy was uploaded into the PMS and the system was rebooted.

The data acquired and transferred by the ES concerning battery SOC, voltage, and current is used by the energy management algorithm to make the real-time calculation of the required battery power. Figure 10c depicts the SOC estimated and sent by the ES during the 5 days of operation. The graph shows how the battery performs approximately one cycle of charge and discharge per day. The good performance of the ES in terms of data reliability and transmission speed is demonstrated by the good performance of the microgrid. Moreover, the variables regarding alarms and warnings are required by the inverter to allow the safe operation of the system. The lack of undesired power limitations or interruptions demonstrate the good performance of the designed ES in the management of alarms and warning events. Finally, the designed GUI that facilitates the interaction between humans, and the battery makes it easy to turn the system on and off; there is also a screen that provides the user with the relevant information about battery status, with variables such as voltage and SOC, as well as warning and alarm events that may be activated due to a malfunctioning of the system. Not only was the performance of the ES suitable for a single second-life battery, but it also operated correctly with an additional first and second generation battery coming from different EVs. Thus, the system presented in this paper is valid for different batteries coming from the same manufacturer and could be easily replicated by other research centers or companies. It should be noted that all the components used are standard commercial products.

## 6. Conclusions

The paper presents an ES architecture that allows continuous data monitoring and control of a repurposed BP and an inverter. To establish the activation process, a number of tests were performed to demonstrate this, starting at the time of the communication between the μC and the BMS to identify the operating parameters, messages, flags, and alarms. Therefore, the aforementioned comments and results demonstrate that our ES is able to ensure battery control and monitoring. This makes it possible to obtain the information regarding its own characterization and, in this respect, to determine whether these BPs can be repurposed without disassembly and rebuilding for use with the INGECON SUN^®^ Storage 1Play inverter in stationary energy storage applications. This reduces ESS investment costs. It has therefore been corroborated that our proposed programming methodology is useful for the development of interfaces focused on second-life ESS management based on Li-ion batteries. For the implementation of our proposal in a second-life self-consumption installation, this study focused its efforts on its improvement as a plug-and-play system based on two aspects. Firstly, the proposed system is able to provide the operators with information on the state of safety, usage, performance, and battery longevity. Secondly, the interface makes it possible to implement a SOC estimator for second-life batteries, described in *A. Soto et al.* [34]., which is a significant issue for the energy management of a second-life battery. 

Finally, it is relevant to mention that our ES proposal can be applied to any other EV Li-ion battery different from the Nissan Leaf^®^ BP to a second-life implementation as stationary ESS, such as Ford Focus^®^, Mini E^®^, Mitsubishi i-Miev^®^, Smart ED^®^, Tesla Roadster^®^, Renault Zoe^®^, BYD F3DM^®^, Opel Ampera^®^, and Toyota Prius^®^ [35]. This is possible due our ES communication protocol is the same used by all the Li-ion battery automotive manufacturers: CAN-BUS. Additionally, to realize this, it is necessary to define in software the message libraries that are established by each BP manufacturer because these messages are not the same from another BP model.

## Figures and Tables

**Figure 1 sensors-22-06376-f001:**
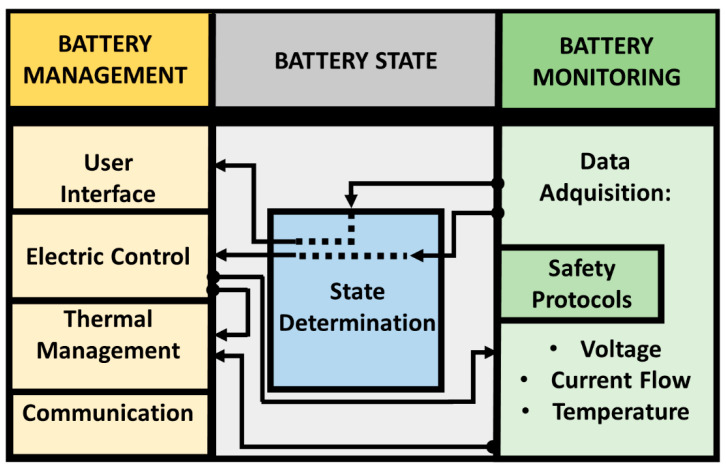
BMS generic architecture.

**Figure 2 sensors-22-06376-f002:**
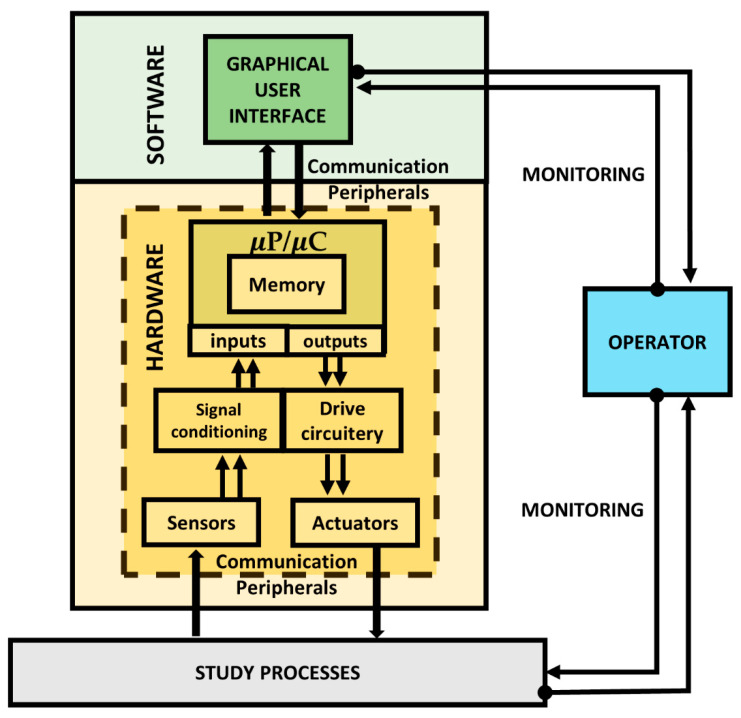
General architecture of an ES.

**Figure 3 sensors-22-06376-f003:**
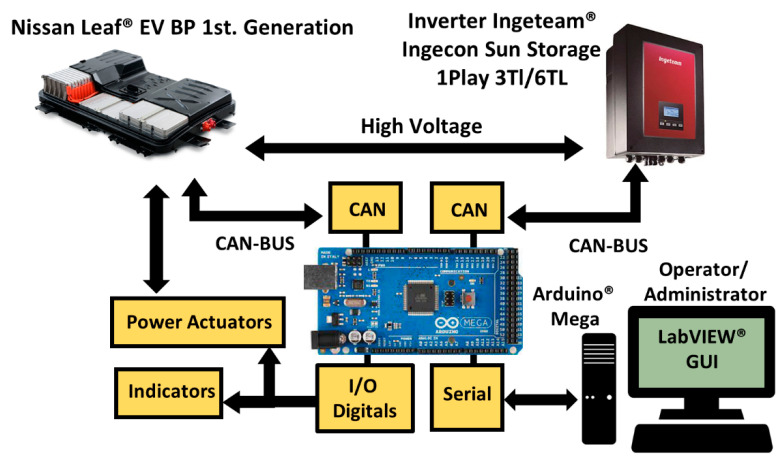
Experimental setup for the EV BP monitoring.

**Figure 4 sensors-22-06376-f004:**
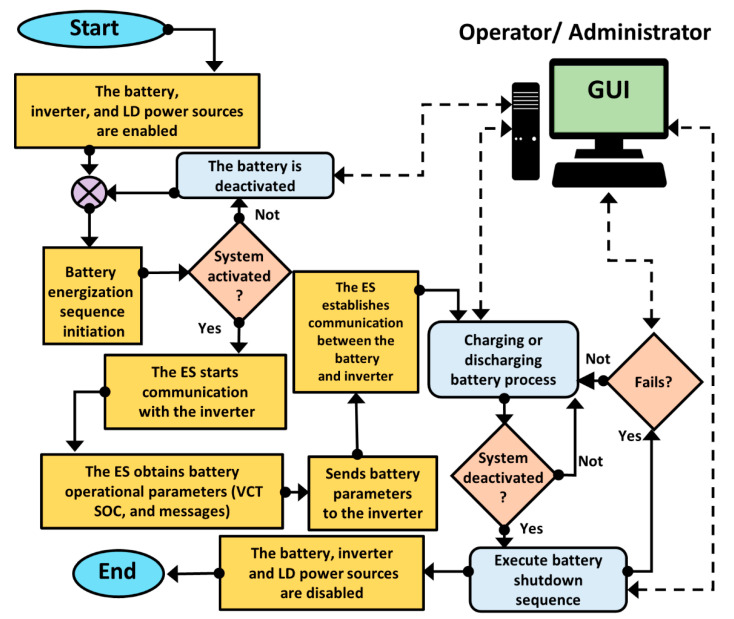
Flowchart describing the second-life ESS work sequence.

**Figure 5 sensors-22-06376-f005:**
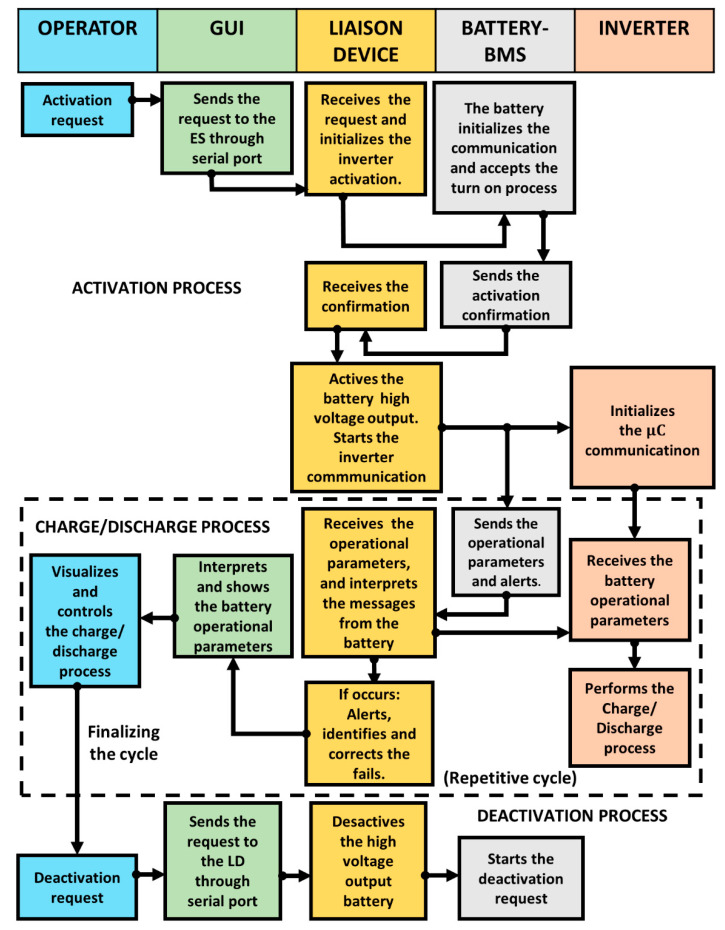
Flowchart describing the *ES* performance.

**Figure 6 sensors-22-06376-f006:**
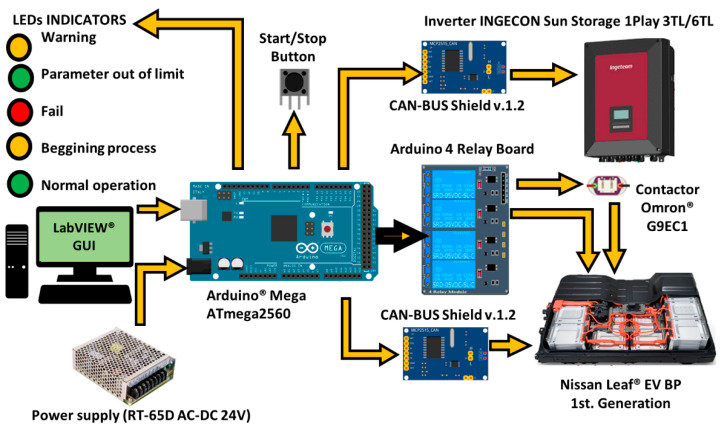
LD hardware architecture.

**Figure 7 sensors-22-06376-f007:**
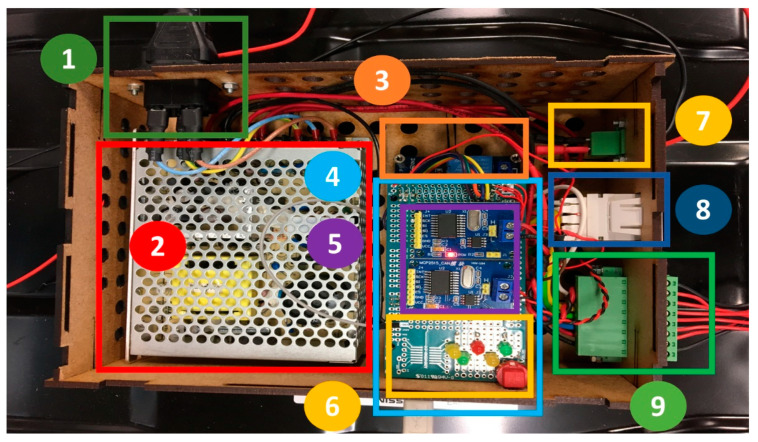
Picture of the LD. 1—Electrical socket; 2—power supply; 3—Arduino 4 Relay Board; 4—Arduino MEGA; 5—two CAN-BUS Shields v.1.2; 6—LEDs and button; 7—HV relay control; 8—CAN inverter communication; and 9—CAN-BUS communication.

**Figure 8 sensors-22-06376-f008:**
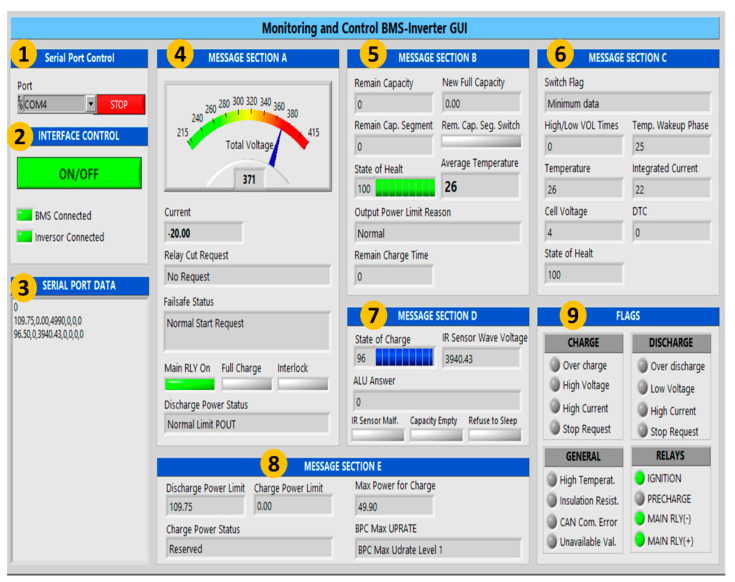
Monitoring and control BMS inverter GUI.

**Figure 9 sensors-22-06376-f009:**
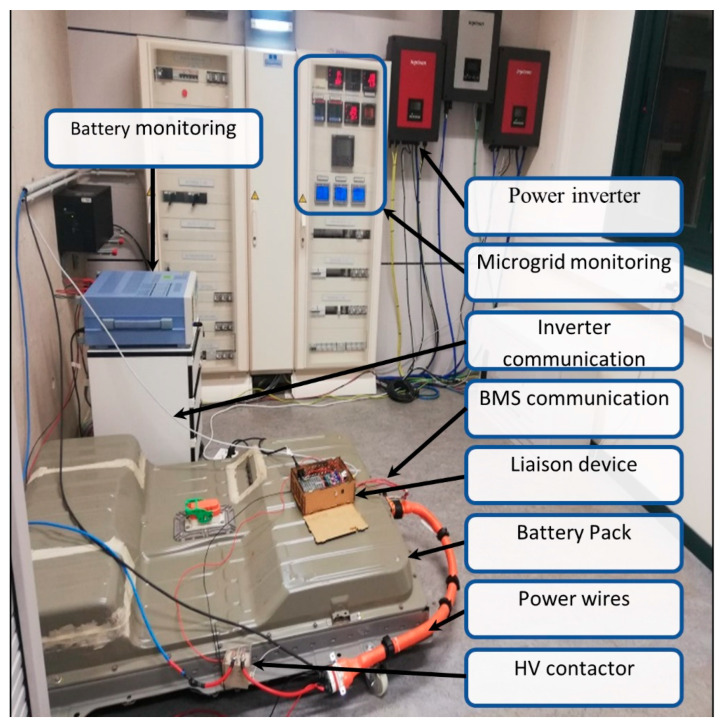
Liaison device connected to the Nissan^®^ EV First Generation Pack at the Laboratory for Energy Storage and Microgrids of the Public University of Navarre.

**Figure 10 sensors-22-06376-f010:**
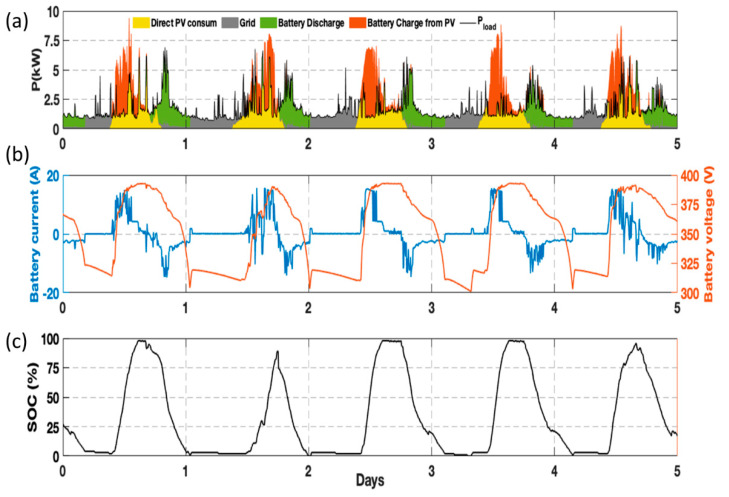
Operation of the microgrid and the battery over five days. (**a**) Energy balance of the microgrid, (**b**) measured battery current and voltage, (**c**) SOC estimated by the ES.

**Table 1 sensors-22-06376-t001:** Main parameters of the battery pack.

	Parameter	Value	Unit
Manufacturer specifications	Nominal capacity	66.2	Ah
	Nominal energy	24	kWh
	Nominal voltage	360	V
	Maximum voltage	403.2	V
	Minimum voltage	240	V
	Weight	293	kg
	Dimensions L × W × H	1570.5 × 1188 × 264.9	mm
	Modules in serial connection (2s2p)	48	-
Measured parameters	Current capacity	38.8	Ah
	Current energy	14.2	kWh
	Energy efficiency	96	%
	Coulombic efficiency	99.7	%
	SOH	58.6	%

**Table 2 sensors-22-06376-t002:** Communication processes between the BMS and Li-ion battery.

Stage	Process	Steps	Description
1	Activation	i. Energize the BMS. ii. Initialize communication with the BMS iii. If the BMS cannot be turned on, the process ends. iv. If the BMS can be turned on, the high voltage output is then available. v. If the BMS sends an alert message, this must be interpreted to guarantee the safety of the device. vi. Initialize the charge/discharge processes. vii. The process ends.	During the process to turn on the battery, the embedded system is able to control the BMS and to comply with the current process. The battery is ready to perform the charge/discharge process.
2	Charge/Discharge	i. Connect the battery and inverter high voltage outputs. ii. Initialize communication between the battery and inverter. iii. Obtain the battery operating parameters (VCT, SOC, and messages). iv. Send these parameters to the inverter. v. Read the inverter current state. vi. Repeat steps iii, iv, and v while the BMS is working or until it is deactivated.	In this process, the battery is available to be charged/discharged. The BMS periodically sends the operating parameters to the embedded system and to the inverter.
3	Deactivation	i. The process initializes. ii. The high voltage output is deactivated. iii. The BMS power supply is deactivated.	The BMS outputs high voltage and the power supply is deactivated.

**Table 3 sensors-22-06376-t003:** Main parameters of the microgrid for different scenarios.

	Without PV and ESS	Only with PV	With PV and ESS
Peak power absorbed from the grid (kW)	6.9	6.9	5.1
Energy consumed from the grid (kWh)	235.5	149.7	78.3
Self-consumption ratio (%)	-	36.4	66.7

## Data Availability

Not applicable.

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
