# Peer review of "Design and On-Field Validation of an Embedded System for Monitoring Second-Life Electric Vehicle Lithium-Ion Batteries"

_sensors, 2022, doi:10.3390/s22176376_

Round 1

Reviewer 1 Report

The embedded system for monitoring second-life electric vehicle lithium-ion batteries proposed in this paper improves battery utilization to a certain extent, reduces battery production costs and reduces environmental pollution. Here are some suggestions for improving the quality of your paper.

1. For battery monitoring system design and applications, it is recommended that you expand more background.

2. Mentioned in the literature review“Ch. Hommalai et al. designed an experimental setup based on CAN protocol to establish communication between the li‐ion battery and a PLC for the purpose of monitoring the dead battery cells.”, the same is the battery monitoring system, please explain the advantages of the method proposed in this article compared with this method.

3. In Section 4 it is mentioned "The inverter controls the power balance of the residential microgrid, depending on the current state of the grid, determining whether it is necessary to use the battery pack as either a power supply or storage device." Then, Please state the reference standard for the battery pack as a power source or storage device.

4. In Section 4 it is mentioned “It is relevant to mention that, when is realized the charging or discharging processes, if occurs any hazardous electrical or thermal battery situation, the operators will be notified by the GUI, and the energy storage system will be automatically deactivated.”, there is no result or data validation for this case in the experiment. Please verify and explain this situation.

5. Please explain "The lack of undesired power limitations or interruptions demonstrate the good performance of the designed embedded system in the management of alarm and warning events."

6. There is no GUI interface picture in the analysis of the results of the field experiment, please add it and give an explanation accordingly.

7. The experiments lack comparisons and the selected experimental objects are too single. The analysis of the experimental results needs to explain the advantages of the method proposed in this paper in more detail.

8. This article only exemplifies the Nissan® LEAF lithium‐ion battery. Please discuss whether batteries from other manufacturers can also be applied to the system designed in this article.

Reviewer 2 Report

Overall, the motivation for the work is well-explained and makes a valid point, electric vehicle batteries are not suitable anymore when their capacity drops by 20% from the nominal value.   Still, the paper requires some English language and style editing, and some passages require further details. That said, the following comments will try to help as much as possible. They are suggestions and are not meant to be disrespectful. The comments are sorted by line number and not by relevance.
But first, a note about acronyms.  

# Acronyms

The paper is inconsistent with its acronyms; one prominent example is Lithium-Ion batteries. The paper starts by referring to them with the acronym Lithium-Ion Batteries (LIBs), line 17-18. Then, at line 41 it uses lithium-ion (li-ion). From there on, it interchangeably uses LIBs and li-ion. I would suggest sticking to the well-known way of writing lithium-ion, which is `Li-ion`.
``` % Required for glossary, in the preamble. \usepackage{glossaries} % Declaration. \newacronym{lion}{Li-ion}{Lithium-ion} % Usage after begin{document} \gls{lion}, or \glspl{lion} for plural ```
Furthermore, sometimes the paper uses the full name and others the acronym. I suggest relying on the glossaries package if the paper is generated from latex. Follow an example:The same applies to microprocessor and microcontroller. The acronym appears to be different from lines 67 and 158. I would suggest writing microcontroller without acronym at line 67 and then starting to define and use the acronym at line 158.
A secondary note: I would suggest using Human Machine Interface (HMI) instead of Man-Machine Interface (M-MI) to have a more gender-neutral and contemporary term.
Follows a list of detailed comments.

# Detailed Comments

## Line 62

`There is therefore` -> `Therefore, there is`  

## Line 71

`The GUI development as a Man-Machine Interface (M-MI) allows the user` -> `The GUI act as a Man-Machine Interface (M-MI), providing the user with`  

## Line 103

`circuits, integrating` -> `circuits, and integrate`  

## Line 109

`Our proposal presented has` -> `Our system has`  

## Line 121

`The charge and discharge processes result in` -> `Charge and discharge operations leads to`  

## Line 122

`original` -> `nominal`  

## Line 123

`Considering,` -> `All considering,`  

## Line 131

`, this device is the` -> `, and the device that performs these operations is the`  

## Lines 134-139

The following sentence is too long, maybe it can be split:

`Its principal task is to ... the LIB [22, 23].` -> `Its principal task is to ensure that the battery operates within safe limits and achieves optimal performance over its useful life under various operating and environmental conditions. Specifically, it needs to keep the temperature stable and the electrical operating variables within the predetermined limits to ensure that the system delivers the energy demanded at any given time and to guarantee the health of the LIB [22, 23].`  

## Line 147-148

To be honest, I never seen text inside an acronym parentheses, it might be allowed. Nevertheless, I would suggest re-writing it to avoid distracting the reader: `Using the CAN communication protocol, the BMS communicates with the Vehicle Management System (VMS), which controls all the vehicle parameters and processes.`  

## Line 149

`automaker` -> `car manufacturer`   Although, `automaker` is in the vocabulary, but some sections later `manufacturer` is used.  

## Line 152

`establishes two-way communication` -> `establishes the two-way communication`

## Line 169

`memory. [24].` -> `memory [24].`

## Lines 172-175

The direct quotation from the work of P. Zhan, should go in between "quotation marks" Alternatively, if the paper is generated by using latex, inside a quote environment (not the quotation one, which is for multi-pharagraph quations).

## Line 186

Remove `, respectively`

## Line 189

`liaison device (LD)` -> `Liaison Device (LD)`
Furthermore, since the acronym LD is defined, shouldn't it be used in the rest of the paper instead of writing liaison device every time? Alternatively, the acronym definition can be removed.

## Line 192

`battery and the` -> `battery, and the`

## Lines 214-215

Suggested rephrasing: `Therefore, the inverter via the liaison device will establish communication with the battery pack through its internal commands.` --> `The inverter will establish a communication channel via the liaison device and its internal commands with the battery pack.`

## Line 219

`liasison` -> `liaison`

## Line 222

Suggested rephrasing: `These processes will be interpreted by the microcontroller as a translator to the inverter.` --> `The microcontroller will interpret these processes, as a translator, and send them to the inverter.`

## Line 234

`begins considering that` -> `begins by considering that`

## Line 235

Remove `the authors`

## Line 236

`capable to manage and establish the` -> `capable of managing and establishing the`

## Line 240

`The first process to attend is to verify` -> `The first task is to verify`

## Line 242

`Once it has been realized the communication` -> `Once the communication is established,`

## Line 244

`Afterwards, the second task to attend` -> `The second task`

## Line 245

`, it which, depends` -> `, which depends`

## Line 247

Suggested rephrasing: `After that, is possible to execute the battery deactivation process whenever the operator wants, occurring the disabling the battery, inverter, and ES power supplies.` --> `Afterward, the operators can execute the battery deactivation process whenever they want, disabling the battery, inverter, and ES power supplies.`

## Line 248

Suggested rephrasing: `It is relevant to mention that, when is realized the charging or discharging processes, if occurs any hazardous electrical or thermal battery situation, the operators will be notified by the GUI, and the energy storage system will be automatically deactivated.` --> `It is worth mentioning that the GUI will notify the operators and automatically deactivate the energy storage system if any hazardous electrical or thermal battery situation occurs during charging or discharging processes.`

## Line 256

`In this subsection, we the authors explain the embedded system overall performance that constitute the second-life energy storage system.`
What does `performance` mean? Maybe it was meant to be `design`, or `architecture`? Probably `architecture` is more in line with what the section describe. Furthermore, beware the missin third-persion `s` it should be `constitutes`, and `we` means the `the autors` no need to write `we the autors`. That said, I would suggest to rephrase the first sentence as follows:
`This section describes the embedded system's architecture that constitutes the second-life energy storage system.`

## Line 257

`In the Figure 5,` -> `In Figure 5,`
No need for the article `the`. The article might be needed when writing somthing like: `In the figure below` or `as shown in the previous figure`.

## Line 261

`capable of visualizes,` -> `capable of visualizing,`

## Line 261

`and controls the processess` -> `and controlling the processess`

## Line 264

The following sentence should be clarified: `as a sequencial machine capable to realize parelells tasks` I can't understand the reason for this sentence and its meaning. Furthermore, it is `sequential`, not `sequencial`.

## Line 265

The following sentence must be clarified: `interpreting the messeges between the BMS and inverter`
So, the MCU receives a message from an entity and relays that message to a target entity, but only after re-formatting the message to be compatible with the target. Is that what the sentence means?

## Line 270-272

There are several things to discuss here, specifically, some elements in this sentence are strange.
   `decides which task to realize`
What does `realize` means? Tasks are usually `executed` or `performed`.
   `there are not any alerts presence by the BMS`
So, does this segment means there are no alerts coming from the BMS, thus it is safe to execute the given task (charge/discharge?
Second, based on my undestanding of the sentence, I suggest to rephrase it: `In real-time the operator by the GUI is informed about the operational parameters current state and decides which task to realize: Charge or Discharge, considering that, there are not any alerts presence by the BMS.` --> `The GUI informs the operator in real-time about the operational parameter's current state and decides which task to execute, either Charge or Discharge, if and only if no alerts are coming from the BMS.`

## Line 272

Given the rephrasing of the previous sentence and given that also the following one is not that clear, I suggest another rephrasing:
`If exist the presence of alerts, the GUI informs to the operator the battery alerts, these can be due undesired scenarios such as low-or-high voltage/current flow, presence of overheating.` --> `These alerts, shown to the operator through the GUI, can be caused by low or high voltage/current flow readings, or the presence of overheating.`
Does the battery also raise alerts with low-temperature readings? Also, some batteries raise alerts when encountering anomalous midpoint voltage deviation. Usually, anomalous midpoint deviations can result in overcharging or undercharging.

## Line 275

`the current state, being able to start` -> `the current state, and based on those can decide whether to start`

## Line 285

`the the` -> `the`
the entire sentence could be rephrased to make it more clear:
`We will now go on to explain the the liaison plug-and-play device, this technology allows the data acquisition from the BMS directly without the presence of any external devices or sensors.` --> `This section describes the liaison plug-and-play device, which allows acquiring data directly from the BMS without needing external devices or sensors.`

## Line 287

`has been realized accomplished`, just keep either `realized` or `accomplished`, even though I would suggest a rephrasing to `This device has been designed based on the manufacturer's technical specifications.`

## Line 289

`Figure 7 presents the interface developed, indicating its components ...` -> `Figure 7 depicts the developed interface.`
Move the list of components in the caption. That way the reader won't need to switch between the text and the picture to understand what they are looking at.

## Line 293

Let us analyze the following sentence:
`The liaison device hardware architecture description begins by mentioning that there is one electrical socket to energize the internal power supply (RT-65D AC-DC 24V) dedicated to energize the internal interface devices and to control a contactor.`
The initial part can be completely removed, and start directly from: `There is one electrical socket to energize ...`.
That said, from the sentence it seems like that the electrical socket is energizing the power supply, and the power supply is controlling the contactor. Which I suppose is not the case.
Maybe, this sentence could be rewritten to make it more clearer, like:
`The electrical socket provides power to the internal power supply (i.e., RT-65D AC-DC 24V), which transforms the outlet voltage level to one compatible with the liaison interface (e.g., 5V).`

## Line 301

`This task starts`, what task are we talking about?

## Lines 306-313

Some things must be clarified about the architecture controlling the contactor. Reasonably, the paper uses a 2-level switch, with the Arduino controlling the relay, which controls the contactor. Relays are meant for circuits with low voltage and current rates, and contactors for high voltage and current rates.
Now, few suggestions and questions:  - Explaining the reasons for the 2-level switch would undoubtedly be helpful for any reader.  - Furthermore, does the circuit need noise suppression across the relay coils(like an RC snubber network or a varistor)?  - If the contactor is inside (or partially inside) the box, switching between 275V/400V and zero, wouldn't that cause more noise?

## Line 319

Suggested rephrasing: `Finally in this section, we are going to present the GUI designed and implemented.` -> `This section describes the GUI and its functionalities.`

## Line 320

`the data flow sent` -> `the data sent`

## Line 322

`battery charge/discharge processes.` -> `battery charge/discharge processes through the GUI.`

## Lines 322-323

Suggested rephrasing: `The data flow communication from the hardware-interface to the GUI is two-way by USB.` -> `The two-way communication from the hardware interface to the GUI relies on a USB cable.`

## Line 323

`shows the GUI, note that the controls and` -> `shows the GUI. Controls and`

## Line 325

`message names, for practical` -> `message names. For practical`

## Line 329

`to the PC, the Stop Button` -> `to the PC, while the Stop Button`

## Line 332

Suggested rephrasing: `Also, in this section is possible to know if the BMS and inverter are connected and communicated with the microcontroller.` -> `This section also shows the BMS and inverter connection status and if they are currently communicating with the microcontroller.`

## Lines 421-423

The opening of conclusions might be clarified and made more fluid:
`In this section we present our conclusions with regard to the embedded system developed. The study permits the continuous data monitoring and control of the battery and inverter.` --> `The paper presents an embedded system architecture that allows continuous data monitoring and control of a repurposed battery pack and an inverter.`

## Lines 470-471

I have noticed that in the list of acronyms, both LIB and LIBs appear. I assume the paper is generated from latex and that the acronyms were produced by using the \gls command.
In general, in LaTeX, you can produce singular forms with `\gls{...}`, or plural forms with `\glspl{...}`. By default, `glspl` will generate the plural form by appending an "s" to the singular form. There are cases where you need to specify the plural form yourself, like with `Frame per Second (FPS)`, which should be `Frames per Second` and not `Frames per Seconds`. You can specify the plural form as follows:
`\newacronym[longplural={Frames per Second}]{fpsLabel}{FPS}{Frame per Second}`
That way, just the singular form will appear in the `List of Acronyms`.

Reviewer 3 Report

The article entitled “Design and on‐field validation of an embedded system for monitoring second‐life electric vehicle lithium‐ion batteries” can be accepted for the publication after minor revision. Please find my comments below.

1.    1 The author did not mention the lithium-based battery's weight capacity for modeling data in mAh (milliampere) or F/g.

2.     2 The author should tabulate tested data in a table compared with energy and power density.

3.      3The authors did not show the degradation rate, including temperature, charge and discharge voltage, current, and the level at which the battery is charged or discharged.

4.      4Temperature parameter (-70 0C to 50 0C) should be also include.

5.    5  Long-term experimental results for stability should be also incorporated.

6.      6Internet of Things (IoT) should be applied to deploy a real-time monitoring system for a Lithium-ion battery.

7.      7State of charge (SOC) Battery's state of health should be discussed under these parameters overcharge, over-discharge, and mileage anxiety of electric vehicle power battery.

8.      8 The conclusion of the investigations should be very much précised and focused.

Round 2

Reviewer 1 Report

After the first round of revisions, the quality of the article has been significantly improved. Below are a few suggestions to further enrich the article.

1. Picture 10 is a little blurry, please adjust the clarity.

2. It is suggested to expand some content on the discussion of simulation results.

Reviewer 2 Report

After reading the new version of the paper, I can assess that the reading is 

much more fluid. All the concepts are adequately introduced, broadening 

this paper's audience spectrum. Some minor things could still be 

improved, but this version is way better.

#1

I noticed the `no` was removed from the following sentence: "The

 GUI informs the operator in real-time about the operational parameter's

 current state and decides which task to execute, either Charge or 

Discharge, if and only if alerts are coming from the BMS."

In the current version, it seems like the GUI only decides what to do if 

alerts come from the BMS. Furthermore, the current version of the 

sentence implies that the GUI makes the decisions, but the user makes 

them. As explained in a subsequent sentence: "The operator receives 

real-time information about the current state, and based on those can 

decide whether to start."

I apologize for not noticing this thing the first time I came across the sentence. I might suggest a last rephrasing: "The GUI informs the operator in real-time about the current state of operational parameters. Based on that information, the operator decides which task to execute, either Charge or Discharge, if and only if no alerts are coming from the BMS."

#2

I suggest avoiding using an abbreviation at line 329: (see below in Fig. 9) -> (see below in Figure 9)

#3

At Line 389, I suggest a rephrasing:

"The Figure 8 also serves as a demonstration of monitoring and control BMS-Inverter GUI in real-time focusing to a battery discharge test considering a current-flow constant."

-->

"Figure 8 depicts the GUI for real-time monitoring and control of BMS-Inverters, focusing on a battery discharge test considering a current-flow constant."

#4

Last thing, could the quality of pictures be improved? In almost all figures, the text is hard to read.
